# Machine-Learning-Assisted Analysis of TCR Profiling Data Unveils Cross-Reactivity between SARS-CoV-2 and a Wide Spectrum of Pathogens and Other Diseases

**DOI:** 10.3390/biology11101531

**Published:** 2022-10-19

**Authors:** Georgios K. Georgakilas, Achilleas P. Galanopoulos, Zafeiris Tsinaris, Maria Kyritsi, Varvara A. Mouchtouri, Matthaios Speletas, Christos Hadjichristodoulou

**Affiliations:** 1Laboratory of Hygiene and Epidemiology, Faculty of Medicine, University of Thessaly, 41222 Larisa, Greece; 2Laboratory of Genetics, Department of Biology, University of Patras, 26500 Patras, Greece; 3Department of Immunology & Histocompatibility, Faculty of Medicine, University of Thessaly, 41500 Larisa, Greece

**Keywords:** COVID-19, T cell receptor, pathogens, diseases, cross-reactivity phenomenon, Machine Learning, MIRA dataset

## Abstract

**Simple Summary:**

For the last two years, COVID-19 has been rigorously studied aiming to identify novel prognostic and therapeutic avenues. Recently, T cell receptor profiling has emerged as a method to associate adaptive immunity with COVID-19 progression and severity. Such data are typically analyzed to explore T cell receptor properties and characteristics in the context of SARS-CoV-2 infection. However, the equally informative alternative analytic strategy of identifying any preferential recognition of viral antigens by the T-cell-mediated immune response is mostly overlooked. In this study, we propose a novel Machine-Learning-oriented approach for analyzing T cell receptor repertoires that is based on the concept of utilising the level at which each SARS-CoV-2 antigen is recognised by the available T cell receptors in each sample from COVID-19-convalescent and healthy cohorts. This approach also allowed us to observe a group of T cell receptors capable of recognising SARS-CoV-2 antigens that were already established in samples from the healthy cohort, leading us to the cross-reactivity phenomenon hypothesis. To explore this, all T cell receptors were examined for being able to recognise antigens from other pathogens and diseases, unveiling evidence of putative cross-reactivity with *M. tuberculosis* and Influenza virus, among others.

**Abstract:**

During the last two years, the emergence of SARS-CoV-2 has led to millions of deaths worldwide, with a devastating socio-economic impact on a global scale. The scientific community’s focus has recently shifted towards the association of the T cell immunological repertoire with COVID-19 progression and severity, by utilising T cell receptor sequencing (TCR-Seq) assays. The Multiplexed Identification of T cell Receptor Antigen (MIRA) dataset, which is a subset of the immunoACCESS study, provides thousands of TCRs that can specifically recognise SARS-CoV-2 epitopes. Our study proposes a novel Machine Learning (ML)-assisted approach for analysing TCR-Seq data from the antigens’ point of view, with the ability to unveil key antigens that can accurately distinguish between MIRA COVID-19-convalescent and healthy individuals based on differences in the triggered immune response. Some SARS-CoV-2 antigens were found to exhibit equal levels of recognition by MIRA TCRs in both convalescent and healthy cohorts, leading to the assumption of putative cross-reactivity between SARS-CoV-2 and other infectious agents. This hypothesis was tested by combining MIRA with other public TCR profiling repositories that host assays and sequencing data concerning a plethora of pathogens. Our study provides evidence regarding putative cross-reactivity between SARS-CoV-2 and a wide spectrum of pathogens and diseases, with *M. tuberculosis* and Influenza virus exhibiting the highest levels of cross-reactivity. These results can potentially shift the emphasis of immunological studies towards an increased application of TCR profiling assays that have the potential to uncover key mechanisms of cell-mediated immune response against pathogens and diseases.

## 1. Introduction

Since SARS-CoV-2 was initially reported in Wuhan, China, there have been 515 million confirmed cases and 6.2 million deaths worldwide as of May 2022, according to the Johns Hopkins Coronavirus Resource Centre [1]. Individuals infected with SARS-CoV-2 exhibit a wide spectrum of responses, from asymptomatic to requiring admission to an intensive care unit [2]. Both the research community and pharmaceutical industry have been rigorously studying COVID-19 [3] and implications of SARS-CoV-2 infection [4], aiming to identify novel prognostic and therapeutic avenues [5,6,7], while also exerting a massive effort to bring a plethora of vaccination schemes to the public within a very limited timeframe [8].

During the past year, there has been a shift in published research highlighting the need to better understand the T cell immunological profile association with COVID-19 progression and severity [9,10,11,12,13,14,15]. T cell immunity appears to be a much more sensitive indicator of past infections in comparison with antibody response. High-throughput methods approaching T cell response can be informative by correlating concepts of clonal depth, breadth and dynamics with symptoms and disease severity [16]. Furthermore, previous studies associated with other viruses such as Middle East Respiratory Syndrome (MERS) and SARS-CoV-1 indicate that coronavirus-specific T cells appear to have long term persistence [17,18]. The same phenomenon also seems to take place in SARS-CoV-2 biology [19,20]. These observations could shape the hypothesis of cross-reactivity between different pathogens, where past infection or vaccination could be protective through long-lived T cell clones. The cross-reactivity phenomenon between SARS-CoV-2 and other coronaviruses has been reported in the literature. Neutralising antibodies isolated from memory-B cells of a SARS-CoV-1 infected individual have been described to react with SARS-CoV-2 surface glycoprotein [21]. In addition, cross-reactive T cells recognising SARS-CoV-2 seem to be acquired during previous infections by other human coronaviruses in 20% to 50% of unexposed individuals around the world [22,23,24,25,26,27]. Those preexisting cells may affect the clinical manifestations of COVID-19 infection.

The adaptive immune response to pathogenic infections is largely dependent on the CD4+ and CD8+ T cell subfamilies [28]. Upon activation, CD8+ T cells can exterminate infected cells and form the long-term memory T cell subpopulation. Conversely, CD4+ T cells control the function of myeloid cells, support CD8+ response and play a key role in the selection of antigen-specific B cells which contribute to the host organism’s neutralising antibody arsenal. T cell receptors (TCRs) are proteins localised on the surface of T cells that are products of recombined genomic sequences during the T cell developmental process. The uniqueness of each TCR sequence essentially controls the T cells’ specificity. TCRs recognise peptides presented by the major histocompatibility complex (MHC) on the surface of most cell types (MHC class-I recognised by CD8-expressing T cytotoxic cells), or on the surface of antigen-presenting cells (APC) (MHC class-II recognised by CD4-expressing T helper cells). The ability of TCRs to recognise more than one peptide-MHC structure defines cross-reactivity [29]. The cross-reactivity of T cells is considered as one arm of the well-described heterologous immunity, namely the immunity that can develop towards one pathogen after exposure to non-identical pathogens [30]. The other arm concerns the bystander activation of T cells, that can be caused by independent activation via released cytokines, or by low-affinity recognition of pMHC [29,31]. Heterologous immunity has been well established in viral infections in several animal models, but also in human viral infections where cross-reactivity of T cells could potentially influence the protection or severity of virus-associated immunopathology [32,33,34].

T cell cross-reactivity, however, comes at a cost. Pathogen-induced autoimmune disorders may also result from cross-reactive T cells, following the immune system’s initial reaction to the pathogen. This phenomenon has been termed molecular mimicry, where peptides derived from pathogens can activate autoreactive T cells due to structural similarity between pathogenic and self-peptides, causing autoimmune diseases or accelerating a previously initiated autoimmune process [35,36,37]. The mechanics of T cells cross-reactivity involves changes in complementarity-determining region (CDR) loop conformation, altered TCR docking on the pMHC, flexible changes in pMHC and structural degeneracy [29,38,39,40]. The binding of TCR with peptides differs concerning their affinity, since the bound peptides could be considerably different in their chemistry [38,39]. In this context, several previous studies have reported the presence of cross-reactivity among different viruses, or epitopes derived from the same pathogen [41,42,43].

Recent evidence in the literature highlights [11,12,13,15] an effort of the research community to explore the TCR repertoire in the context of several levels of COVID-19 infection severity, by utilising data from high-throughput TCR-Sequencing (TCR-Seq) assays. The majority of the work focuses on developing computational methods to unveil differences and similarities between healthy and infected subjects related to the TCR repertoire diversity, CDR3 length distribution and the V and J gene segment preference [11,12,13,14]. Other studies have attempted to combine the aforementioned TCR-related statistics with Machine Learning (ML), aiming to provide predictive tools to distinguish healthy and infected subjects [44,45]. To the best of our knowledge, however, no studies exist in the literature which attempt to approach this field from the viewpoint of the extent each SARS-CoV-2 antigen is recognised by the TCR repertoire of COVID-19 infected and non-infected individuals.

In this study, we propose a novel ML-based approach for analysing TCR repertoires derived from the Multiplexed Identification of T cell Receptor Antigen (MIRA) specificity assay [46] (Figure 1A,B). Such data have been recently generated through academic partnerships with the industrial sector, and were released in the form of freely accessible databases such as immunoACCESS© [15]. This resource includes the immunoSEQ dataset of sequenced TCR beta chain (TCRb) repertoires from COVID-19-exposed, -infected or -recovered individuals who have participated in the Immune Response Action to COVID-19 Events study, as well as thousands of patients’ blood samples collected by international institutions globally. The immunoACCESS© repository also includes the MIRA dataset which is complementary to immunoSEQ; the repository catalogues TCRb sequences and TCR specific information about the peptide’s interaction in amino acid level, as well as the targeting epitope molecule it comes from. Our approach (Figure 1B) focused on the MIRA dataset and utilised, for the first time, the level at which each SARS-CoV-2 antigen is recognised by the available TCRs in each sample, to train several ML algorithms that can distinguish samples from COVID-19-convalescent and healthy (no known exposure) cohorts with over 85% accuracy. The module for highlighting the importance of each antigen revealed that the TCR clones recognising ORF7b, nucleocapsid phosphoprotein, as well as ORF1ab, ORF10, ORF3a and membrane glycoprotein to a lesser degree, play a key role for the classification task. Additionally, all MIRA TCRs, regardless of their assigned cohort, were further analysed to determine their potential for recognising epitopes originating from pathogens other than SARS-CoV-2 (Figure 1C). To this end, data from public TCR databases were processed to unveil evidence of putative cross-reactivity between SARS-CoV-2 and multiple pathogens and other diseases, with *M. tuberculosis* and Influenza virus being the most cross-reactive.

## 2. Materials and Methods

### 2.1. Data Collection and Pre-Processing

TCRs that are able to bind to SARS-CoV-2 epitopes were retrieved from the immuneACCESS© database [15] (Figure 1A). These SARS-CoV-2 specific TCRs are part of the MIRA dataset which is based on 144 samples (experiments) obtained from cohorts with exposed subjects and healthy controls (Figure 1B and Figure 2A). It should be noted that for certain subjects, more than one sample is available in MIRA. Specifically, 90 samples originate from COVID-19-convalescent, 39 from healthy (no known exposure), 4 from COVID-19-acute, 8 from COVID-19-non-acute and 3 from COVID-19-exposed subjects. Samples from the COVID-19-acute, COVID-19-non-acute and COVID-19-exposed cohorts were excluded from the analyses presented herein, due to their limited number. TCR sequences were initially filtered to keep CDR3 regions delimited by a conserved cysteine at the start and a conserved phenylalanine or tryptophan at the end (anchors of CDR3 region). Unproductive CDR3 segments and sequences containing special characters not corresponding to amino acids (X, #, *, etc.) were also excluded. MIRA was further filtered to keep information associated only with functional V genes, removing information related to pseudogenes and ORFs according to the immunogenetics information system (IMGT) [47]. Clonotypes with ambiguous V gene family members (denoted with X) were also removed. The analysis was focused on the remaining 130,072 (120,128 unique) TCRs detected in the studied cohorts (28 healthy and 85 convalescent subjects). A total of 1006 unique TCRs originate from the minigene-detail file, 114,411 from peptide-detail-ci (CD8+) and 4651 from peptide-detail-cii (CD4+). In addition, 34 unique TCRs were found in both minigene-detail and peptide-detail-ci files, and 26 TCRs were found in both peptide-detail-ci and peptide-detail-cii files. The unique number of TCRs per 1000 TCRs in each sample is depicted in Figure 2B.

All remaining TCRs were further analysed from the SARS-CoV-2 antigens’ point of view. For every sample, each TCR was assigned to an antigen category (N = 11) based on its epitope recognition ability (Figure 1B and Figure 2C). Some TCRs were able to recognise epitopes from different antigens. For these cases, the assignment to the corresponding antigens was weighted based on the number of antigens. The number of TCRs per antigen was normalised based on the total number of TCRs per sample. This approach enabled the representation of each sample by an 11-dimensional vector and facilitated the aggregation of a dataset used to build several ML models that can classify samples into the convalescent and healthy categories and explore the underlying biology (Figure 1B).

Data catalogues derived from three public TCR databases with immunogenetic information were downloaded to investigate the MIRA dataset’s TCR involvement in immune response during other infections. Pathology-associated TCR database McPAS [48] is a manually curated dataset of TCR sequences associated with various pathological manifestations, containing information about the T cell type, organ or tissue antigen target and related MHC molecules (version 4 January 2022). TCR3d [49] is a structural repertoire database including experimentally determined TCR structures and complexes. It also contains TCR sequences and related antigenic peptides and MHC molecules. We used a data frame derived from TCR3d focused on TCRb CDR3 sequence level and the association with viruses (version 13 January 2022). VDJdb [50] is a database containing TCR sequences, their cognate antigens and related MHC molecules (version 22 March 2022). All three aforementioned databases were filtered to keep information about immune response in human species and CDR3 sequences associated with TCRb; the MIRA filtering strategy described earlier in this section was also applied here. Additionally, in the case of VDJdb, CDR3 sequences with zero confidence score were removed from downstream analyses. As stated in VDJdb’s documentation, the higher the score the more confidence there is in the antigen specificity annotation of a given TCR clonotype. Some VDJdb sequences were processed during fixing steps according to IMGT nomenclature and were included in our analysis. Furthermore, the McPAS sequences associated with antigen identification method id “3” were removed in accordance with the database’s recommendation for confidence in the accuracy of the data. 

### 2.2. Machine Learning Model Training and Feature Importance Estimation

All 90 COVID-19-convalescent samples were labelled as positives, and the 39 healthy (no known exposure) samples as negatives (Figure 1B and Figure 2A). Each sample consists of an 11-dimensional vector of normalised values, for every SARS-CoV-2 antigen, that represents the percentage of TCRs recognising the antigen’s epitopes from the total amount of SARS-CoV-2-specific TCRs in the sample (Figure 1B and Figure 2C). Visualisation of the data in the principal component analysis (PCA) space is depicted in Figure 2C. Principal component loadings can be found in Appendix A. Both positive and negative samples were randomly divided into training and test sets based on a 7:3 ratio. This process was repeated 20 times to generate an equal amount of training/test set combinations and control for any bias that could result from the splitting process (Figure 1B).

These sets were used to train and evaluate a total of 100 models based on popular ML algorithmic families (Figure 1B): Gaussian Naive Bayes (GaussianNB), Decision Trees (DT), K-Nearest Neighbours (KNN), Random Forests (RF) and Support Vector Machines (SVM). The hyperparameters of each model were tuned based on a grid search approach, and balanced accuracy was the target metric for choosing the best performing model. For GaussianNB and the variance smoothing parameter, the grid search was run on the values 1 × 10^−9^, 1 × 10^−8^ and 1 × 10^−7^. For DT, the benchmarked values for maximum depth were 10, 30 and 90 with the maximum features parameter set to None. In the case of KNN, the k-neighbours parameter values were 2, 5 and 10. The algorithmic options for selecting nearest neighbours were ball_tree, kde_tree and brute, and the distance metric parameter values were 1 (manhattan) and 2 (euclidean). For RF, the maximum depth values were 10, 30 and 90; the number of estimators were 10, 50 and 100. The maximum features parameter was set to None. The SVM kernel was set to radial basis function and the different ‘C’ parameter values were 0.1, 1, 10 and 100. The gamma values were 1, 0.1, 0.01 and 0.001. All models were trained based on a 10-fold cross validation scheme repeated 10 times. Subsequently, the best performing model was evaluated on its designated test set.

This approach resulted in the calculation of performance metrics such as balanced accuracy, precision, sensitivity, specificity and NPV (Figure 3A,B). Since 20 models were trained and benchmarked for each ML algorithm, all performance plots depict the metrics’ score distributions from all test sets, providing hints of putative training/test split bias and data heterogeneity. Additionally, the importance of each feature was estimated by repeated (N = 50) random shuffles of single feature values to assess the decrease or increase of the models’ performance (Figure 3C). For each ML algorithm, features with a median score above 0.01 were selected for a second round of training and evaluation, following the previously aforementioned strategy (Figure 3D).

### 2.3. Identification of TCRs That Recognise Epitopes from Antigens of SARS-CoV-2 and Other Pathogens and Diseases

Several definitions of the clonotype concept exist as various studies approach it with different immunogenetic characteristics. The MIRA dataset contains TCRb sequences targeting specific SARS-CoV-2 epitopes and the TCR bioidentity is described by CDR3 amino acid sequence, V and J genes. In this analysis, every clonotype consists of sequences characterised by the same V gene family member, the same J gene family member and the same CDR3 sequence in amino acid level. Clonal expansion of every clonotype in each sample (experiment) was calculated by counting the times it appears divided by the total MIRA clonotypes detected. Hence, clonal expansion was defined as a measure of the T cell proportion expressing a specific TCRb sequence. The mean clonal expansion of each clonotype was assessed by calculating the mean of expansion values from all experiments where this clonotype was detected (Figure 4A).

The six most frequent clonotypes were further analysed from the viewpoint of antigen recognition, clonal expansion range and clinical impact. The level of clonal expansion was also calculated separately for the convalescent and healthy cohorts (Figure 4B). Additionally, each clonotype was characterised for the clinical cohort distribution and statistical significance was also determined with Fisher’s exact test (Figure 4C).

The cross-reactivity phenomenon between SARS-CoV-2 and other pathogens was also examined. We used the public TCR databases McPAS [48], TCR3d [49] and VDJdb [50] to confirm the existence of different pathogens and other diseases associated with clonotypes targeting SARS-CoV-2 epitopes in the MIRA dataset. Quantitative analysis was conducted calculating the number of unique CDR3-mediated connections associated with each pathology to capture the number of putative cross-reactive TCRs (Figure 5A). In certain cases, a single CDR3 sequence can recognise epitopes from multiple antigens. Thus, the number of SARS-CoV-2 antigen connections with other pathogens’ and diseases’ antigens is not equal to the number of unique MIRA CDR3 sequences (Figure 5 and Appendix A). This approach was applied twice, once by screening all CDR3 sequences (Figure 5, Appendix A) from the aforementioned databases (derived from CD8+ and CD4+ T cells) and once by targeting only CD8+ or CD4+ T cells, to examine any potential functionality bias (Appendix A, Appendix A). To identify locations crucial for the interaction with cross-reactive CDR3 regions, the epitopes were aligned to the pathogenic proteins they derive from (Figure 4D,E and Figure 5B). This was the first step to characterise the specific domains and functionality of antigens recognised by the same CDR3 sequences.

It should be noted that these public databases include information for both CD4+ and CD8+ T cells and a data bias exists due to the numerous study results associated with specific pathogenic cases. To briefly mention the most notable cases, there are 16,162 unique CDR3 sequences associated with *M. tuberculosis* (derived from 14,992 CD4+ and 1183 CD8+ T cells), 3639 with Influenza virus (derived from 155 CD4+ and 3479 CD8+ T cells), 2663 with CMV (derived from 2658 CD8+ T cells), 1583 with HIV (derived from 350 CD4+ and 1514 CD8+ T cells) and 1437 with EBV (derived from 1334 CD8+ T cells). The total number of unique CDR3 sequences does not coincide with the CD4+ and CD8+ subsets, as some CDR3 sequences seem to be associated with both subpopulations. In addition, for some CDR3 sequences there is no information about T cell type in the public TCR repositories. However, separating T cell populations reflects important information about the immune response against the studied pathogens.

## 3. Results

### 3.1. MIRA Dataset Exploration Unveils Differentially Recognised SARS-CoV-2 Antigens between Convalescent and Healthy Samples

The MIRA dataset includes 144 samples (experiments) from the immunoACCESS study [15] that are divided into five cohorts: (1) COVID-19-convalescent, (2) healthy (no known exposure), (3) COVID-19-acute, (4) COVID-19-non-acute and (5) COVID-19-exposed (Figure 2A). Most cohorts have a balanced male-to-female ratio; however, there are 26 samples from unknown gender, of which the majority (N = 21) are reported as COVID-19-convalescent cohort. The COVID-19-acute, -non-acute and -exposed cohorts were removed from all subsequent analyses due to low sample numbers. Initial analysis regarding the normalised number of unique TCRs in each sample revealed there is no statistically significant difference between the healthy and convalescent cohorts (Figure 2B).

Most existing studies focus on TCR properties such as the underlying VJ rearranged sequences, CDR3 sequences, CDR3 size and clonal depth/diversity [11,12,13,14]. This information has been frequently used to characterise the TCR repertoire of immune response against COVID-19 and to train ML algorithms that are able to distinguish between samples of distinct cohorts [44,45]. In this study, a different approach was adopted. Rather than using the aforementioned TCR-related data, we exploited the MIRA dataset [15], and the connection between TCRs and the SARS-CoV-2 epitopes to generate an 11-dimensional vector representing the level at which each SARS-CoV-2 antigen is recognised by the TCRs in each sample (Figure 1B and Figure 2C).

The results of this approach revealed that even in samples from the healthy cohort, all SARS-CoV-2 antigens are recognised by TCRs to some extent, suggesting either previous unreported COVID-19 infection of subjects in the healthy cohort, or putative cross-reactivity between SARS-CoV-2 and other pathogens. Interestingly, the ORF1ab and surface glycoprotein are the two antigens recognised by TCRs with the highest overall clonal depth, although the difference in clonal expansion between the two cohorts is not statistically significant. More importantly, the nucleocapsid phosphoprotein, ORF7b, ORF10, ORF8 and envelope exhibit statistically significant differences in the number of TCRs recognising their epitopes between the two cohorts. However, the ORF10, ORF8 and envelope proteins are recognised by TCRs with low clonal depth. The projection of these samples on PCA space (Figure 2D, Appendix A) has led to the assumption that this approach could be used to develop a ML-based framework for specifically distinguishing between samples from the two MIRA cohorts (Figure 1B), aiming to extract additional information from MIRA that possibly cannot be derived from the statistical analysis described above.

### 3.2. Explainable ML Highlights Key SARS-CoV-2 Antigens for Classifying Samples into the Convalescent and Healthy MIRA Cohorts

The strategy of modelling the MIRA dataset involved a repeated process (N = 20) of separately splitting samples from healthy and convalescent cohorts into training and test sets. At each split, five models based on GaussianNB, DT, KNN, RF and SVM were trained and evaluated (Figure 1B).

Using a prediction score cut-off of 0.5 enabled the extraction of performance metrics on each test set (Figure 3A). SVM was the overall best performing algorithm with a median performance of at least 0.75 in all metrics. Notably, the SVM algorithm exhibits 0.856 median balanced accuracy, 0.896 precision, 0.962 sensitivity, 0.75 specificity and 0.9 negative predictive value (NPV). To observe SVM’s performance across the whole spectrum of prediction score thresholds, an incremental cut-off was applied and all metrics were calculated at each step (Figure 3B).

To assess the importance of each feature, a repeated (N = 50) feature value perturbation process was applied for all ML algorithms (Figure 3C). Overall, the most important features are ORF7b, nucleocapsid phosphoprotein as well as ORF1ab, ORF10, ORF3a as well as membrane glycoprotein to a lesser extent. After selecting only the most important features for each algorithm, the training and evaluation process was repeated. This resulted in slightly improved performance for GaussianNB, KNN and SVM algorithms, but not for DT and RF, as expected considering their innate ability to readily rely only on important features (Figure 3D).

### 3.3. Exploratory Analysis of MIRA TCRs Unveils Evidence of Putative Cross-Reactivity between SARS-CoV-2 and Other Pathogens and Diseases

MIRA TCRs exhibit diverse frequency of occurrence and clonal expansion levels (Figure 4A). In general, the most frequent clonotypes in the dataset present low mean expansion after being triggered with the MIRA assay, while the least frequent clonotypes are associated with the highest expansion levels.

The analysis focused on the six most common clonotypes with frequencies of greater than 11% of the total number of subjects; CASSIRSSYEQYF+V19-01+J02-07 (CD8+ T cell TCR in MIRA), CASSLAGAYEQYF+TCRBV05-01+TCRBJ02-07 (CD8+), CASSLSAPQETQYF+TCRBV27-01+TCRBJ02-05 (CD8+), CASSLSSPQETQYF+TCRBV27-01+ TCRBJ02-05 (CD8+), CASSDRGPNQPQHF+TCRBV27-01+TCRBJ01-05 (CD8+) and CASSDRGPTDTQYF+ TCRBV27-01+TCRBJ02-03 (CD8+), that were found in 23%, 15.04%, 13.27%, 12.38%, 11.5% and 11.5% of total MIRA subjects, respectively (Figure 4A, marked with arrows). The distribution of the clonal expansion level in samples belonging to the two cohorts was also calculated, not unveiling any statistically significant differential expansion between the two cohorts (Figure 4B, based on Mann-Whitney; the statistical test could not be performed for some of the TCRs, denoted with *p*-val N/A). For every related sample, the number of times each clonotype appears in the corresponding sample(s) was divided by the total number of the sample’s clonotypes.

Calculation of cohort distribution took place in each clonotype’s subgroup compared to the whole sample with Fisher’s exact test (Figure 4C). We observed a significant difference in the case of the most frequent clonotype CASSIRSSYEQYF+V19-01+J02-07 (*p*-value = 0.0019) and the fourth most common clonotype CASSLSSPQETQYF+V27-01+J02-05 (*p*-value = 0.038). Specifically, CASSIRSSYEQYF+V19-01+J02-07 clonotype is detected in a sample’s subpopulation where most subjects are characterised as healthy. In contrast, CASSLSSPQETQYF+V27-01+J02-05 clonotype is detected only in convalescent subjects.

Additionally, the SARS-CoV-2 antigen targets of the six most common clonotypes were identified. CASSIRSSYEQYF+TCRBV19-01+TCRBJ02-07 interacts with surface glycoprotein and ORF1ab, CASSLAGAYEQYF+TCRBV05-01+TCRBJ02-07 recognises the nucleocapsid phosphoprotein, CASSLSAPQETQYF+TCRBV27-01+TCRBJ02-05 recognises ORF1ab and envelope, and CASSLSSPQETQYF+TCRBV27-01+TCRBJ02-05, CASSDRGPNQPQHF+TCRBV27-01+TCRBJ01-05 and CASSDRGPTDTQYF+TCRBV27-01+TCRBJ02-03 interact with ORF1ab. Public databases such as McPAS, TCR3d and VDJdb were used to query the putatively cross-reactive components of the aforementioned TCRs. CASSIRSSYEQYF+TCRBV19-01+TCRBJ02-07 (catalogued as CD8+ in the MIRA dataset and the other TCR catalogues as well) was also found to interact with Influenza virus and Epstein-Barr Virus (EBV). The description of this specific clonotype as part of the immune response against Influenza virus has been previously described [51]. The most common clonotype, which seems to be cross-reactive according to our results (CASSIRSSYEQYF+V19-01+J02-07), is significantly associated with healthy individuals (*p*-value = 0.0019). The exact opposite phenomenon takes place in CASSLSSPQETQYF+V27-01+J02-05 analysis, where we did not detect any cross-reactivity and a significant correlation was observed with the COVID-19-convalescent cohort (*p*-value = 0.038), shaping the hypothesis of diversification in SARS-CoV-2 immunity dependent on past infections and/or vaccination.

These results verified the suspicions of cross-reactivity that surfaced through observations based on results of the initial analysis presented in Figure 2C. Figure 4D highlights the most common TCR’s recognition sites on surface glycoprotein from SARS-CoV-2 and matrix protein 1 (M1) from Influenza A virus. The same information is also depicted in the form of a circular plot in Figure 4E. The remaining five most common clonotypes (CASSLAGAYEQYF+TCRBV05-01+TCRBJ02-07, CASSLSAPQETQYF+TCRBV27-01+TCRBJ02-05, CASSLSSPQETQYF+TCRBV27-01+ TCRBJ02-05, CASSDRGPNQPQHF+TCRBV27-01+TCRBJ01-05 and CASSDRGPTDTQYF+ TCRBV27-01+TCRBJ02-03) were not found in VDJdb, McPAS or TCR3d to interact with other pathogens.

To further assess the cross-reactive properties of all TCRs in the MIRA dataset, the epitopes in MIRA as well as epitopes in the three aforementioned public TCR databases were used to match MIRA CDR3 sequences with antigens from SARS-CoV-2 and other pathogens and diseases (Figure 5, Appendix A). In general, some CDR3 sequences can recognise epitopes from multiple antigens. Therefore, the number of SARS-CoV-2 antigen connections with other pathogens’ and diseases’ antigens might not be equal to the number of unique MIRA CDR3 sequences. In Figure 5, the connections between antigens from SARS-CoV-2 and other pathogens (Appendix A) and diseases have not been divided according to the CDR3 sequence origin (CD8+ or CD4+ T cell), since we have observed that some sequences are associated with CD8+ T cells in MIRA and CD4+ T cells in McPAS, VDJdb and TCR3d databases, or vice versa. The same analysis was repeated after grouping CDR3 sequences based on their CD8+ or CD4+ T cell origin in both MIRA and external databases (Appendix A, Appendix A).

As shown in Figure 5A and Appendix A, evidence regarding the cross-reactivity phenomenon is widespread and links SARS-CoV-2 to a plethora of pathogens and other diseases that can be grouped into three major categories (Appendix A). The first category includes *M. tuberculosis* and viruses such as Influenza virus, Cytomegalovirus (CMV), EBV, Human Immunodeficiency Virus (HIV), Hepatitis C Virus (HCV), Yellow Fever Virus (YFV), Dengue Virus (DENV) and Human T-lymphotropic Virus type 1 (HTLV-1). From 2136 CDR3 sequence connections between the MIRA and other repositories, 1792 (83.9%) can recognise epitopes from SARS-CoV-2 and members of the first category (Appendix A). The second category consists of malignancies and malignancy-related agents such as Melanoma, Breast Cancer and Neoantigens. Roughly 9.9% (211 out of 2136) of the CDR3-mediated connections between MIRA and the three public TCR profiling databases recognise epitopes from the second category (Appendix A). On the other hand, 6.2% (133 out of 2136) of the connections recognise epitopes from the third category, which reflects auto-immune states and disorders arising from external stimuli such as Celiac Disease, Inflammatory Bowel Disease (IBD), Diabetes Type 1, Psoriatic Arthritis, Allergy and Toxic Epidermal Necrolysis (Appendix A).

The most cross-reactive partner of SARS-CoV-2 was found to be *M. tuberculosis*, with 747 CDR3-mediated connections recognising epitopes from both pathogens (Figure 5A, Appendix A). These CDR3 sequences were isolated mostly from CD8+ T cell TCRs in MIRA and CD4+ in the other TCR repositories (Appendix A). Specifically, 271 connections are associated with ORF1ab and 127 with surface glycoprotein. Influenza virus exhibits the second highest number of CDR3-mediated connections (498 out of 2136) with SARS-CoV-2 (Appendix A). The majority of CDR3 sequences are associated with CD8+ T cell TCRs in both MIRA and the public TCR profiling databases (Appendix A), in contrast to the case of *M. tuberculosis*. Notably, 144 connections are related to surface glycoprotein and 123 to ORF1ab.

To have a complete view of the cross-reactivity phenomenon, we further generated a circular plot depicting the “cross-talk” between distinct antigen regions through the recognition by MIRA CDR3 sequences (Figure 5B, Appendix A). To ensure concise visualisation, a subset with the most cross-reactive pathogens, as depicted in Figure 5A, was selected for generating the plot, including *M. tuberculosis*, Influenza virus (A subtype), CMV, EBV and HIV. Each scaled segment in the circle represents an antigen and every antigen is coloured based on the pathogen it belongs to. The links connecting antigen pairs correspond to the “cross-talk” between the antigens’ segments through their ability to be recognised by a specific CDR3 in MIRA. Although *M. tuberculosis* is the most cross-reactive partner of SARS-CoV-2, the matching epitope information is missing from McPAS, TCR3d and VDJdb for the majority of CDR3 sequences. Thus, the *M. tuberculosis* connections in Figure 5B are severely limited. The same phenomenon was also observed for the other pathogens, but to a lesser degree.

## 4. Discussion

Over the last two years, the global impact of COVID-19 on healthcare [52] and the socio-economic [53] field has been devastating. The response of both the scientific community and pharmaceutical industry was swift and decisive in exploring the biological aspect of SARS-CoV-2 and its pathological implications, as well as delivering pharmaceutical products that could assist in restraining COVID-19. During the past year, an observed shift in the literature was evident, highlighting the T cell immunological profile characterisation in the framework of COVID-19 progression and severity [9,10,11,12,13,14,15]. Collaborations between academia and industry resulted in the publication of immunological datasets from studies with thousands of subjects, such as the immunoACCESS© resource [15]. Specifically, the MIRA dataset provides access to thousands of TCR clonotypes that can specifically recognise SARS-CoV-2 epitopes (Figure 1A).

Our strategy is based on a novel ML-oriented TCR profiling assay analytic approach, which can highlight the targeted antigens of the immune response against pathogens and diseases (Figure 1B and Figure 3). During the last two decades we have experienced an abundance of breakthroughs in biotechnology that facilitated the dawn of the big data era for biology. We believe the scientific community should emphasise the development of efficient and accurate computational approaches, to exploit the wealth of information embedded in the ever-increasing volume of biomedical data. ML can be the ideal substrate for combining data from heterogeneous sources of information, while unveiling higher-order and more abstract connections between the underlying mechanisms of biological phenomena and the environment. In the context of COVID-19 related research and other infectious diseases, ML could be used for combining epidemiological surveillance with data from immunoassays (TCR-Seq and others), genomics, transcriptomics and even metagenomics. Such approaches can provide a solid foundation for understanding the entanglement of genetic factors and the environment, as well as their implications for the progression of pandemics, for example, within different populations.

One key observation in our study is that T cell TCRs in the MIRA dataset with the ability to recognise epitopes from ORF1ab and surface glycoprotein exhibit similar levels of clonal expansion between the two cohorts and present the highest clonal expansion levels in the healthy cohort (Figure 2C); this suggests either previously unreported COVID-19 exposure or putative cross-reactivity between SARS-CoV-2 and other pathogens. The former hypothesis could not be verified by any means, since the samples’ collection date in MIRA is not available. Therefore, we proceeded exploring whether MIRA TCRs exhibit cross-reactive properties that can be a product of an immune response against both SARS-CoV-2 and other pathogens and diseases.

Our analysis was based on the CDR3 sequences that are common between MIRA and McPAS, TCR3d and VDJdb repositories. The results unveiled widespread evidence of putative cross-reactivity that link SARS-CoV-2 to a plethora of pathogens and other diseases (Figure 5), which can be grouped into three major categories: (a) *M. tuberculosis* and viruses including Influenza virus, CMV, EBV and HIV among others, (b) malignancies and malignancy-related agents, and (c) auto-immune states and disorders. Interestingly, the majority of CDR3 sequences that target pathogens in the first category originate from CD8+ T Cell TCRs, according to McPAS, TCR3d and VDJdb, with *M. tuberculosis* being the exception. The association of BCG vaccine with CD4+ T cell response against *M. tuberculosis* has been previously reported in the literature [54]. In contrast to the first category, we observed the exact opposite pattern for CDR3 sequences that are associated with pathological states from the second and third categories, since they are derived mostly from CD4+ T cell TCRs.

*M. tuberculosis* was found to exhibit the highest levels of T cell cross-reactivity with SARS-CoV-2. These results are in accordance with published epidemiological studies conducted prior to SARS-CoV-2 vaccine implementation, which have suggested a negative association between incidence, morbidity and mortality of COVID-19 and national Bacille Calmette–Guérin (BCG) vaccination programs. Specifically, in countries where national BCG vaccination has been implemented, lower numbers COVID-19 cases and related deaths have been recorded [55,56,57]. A study conducted by Escobar et al. [58] found a 10.4% reduction in mortality from COVID-19 for every 10% increase in a country’s BCG index. Discovered in 1921, even today the function of BCG vaccine remains obscure to a certain extent. The BCG vaccine contains attenuated Mycobacterium bovis, and induces humoral and adaptive immunity, activating both non-specific and cross-reactive immune responses in the host [59,60] against a variety of infectious (viruses, bacteria, fungi and parasites) and non-infectious agents. Epigenetic and metabolic reprogramming of innate immune cells, known as trained immunity, is considered responsible for these protective effects [61,62,63,64].

Influenza virus exhibits the second highest number of CDR3 sequences that are putatively cross-reactive with SARS-CoV-2. This type of cross-reactivity could be attributed to the seasonal vaccination against Influenza virus and relevant exposure of a large portion of the population to this virus. The role of protective immunity induced by the polyvalent influenza virus vaccine (against Influenza A virus and/or Influenza B virus subtypes) and the likelihood of COVID-19 has been previously examined [65,66,67]; meanwhile, others explored this association from a clinical manifestation and disease outcome perspective [68]. Although the precise pathophysiological mechanisms underlying this association require further investigation, three main theories have been put forward. The first theory relates to antigenic mimicry which results in clonal activation and lymphocytes proliferation [69]. Depending on each individual’s HLAs, only a limited number of epitopes can be recognised, and those are the immunodominant ones. A second theory of trained immunity has also been proposed as the mechanism behind these beneficial heterologous effects of vaccines [70]. Influenza virus vaccination acts as a non-specific exciter of our immune response [71]. Debisarun et al. found that rather than binding, cytokines broaden T cell responses against SARS-CoV-2 [72]. Salem et al. suggested flu-induced bystander immune response as a probable protective mechanism [73].

Information related to T cell cross-reactivity between SARS-CoV-2 and the remaining viruses from the first category of cross-reactive pathogens is scarce in the literature. Cellular cross-reactivity against EBV and SARS-CoV-2 has not been studied to date. However, there have been reports associating the clinical manifestation of COVID-19 with reactivation of EBV infection and correlating it with severe disease progression, thus underpinning a possible entanglement [74,75,76]. Cross-reactivity between SARS-CoV-2 and HIV has also been reported in studies describing false positive HIV results in COVID-19 patients. Such cases were associated with antibody cross-reaction during immunoassay screening tests, while cross-reactive CDR3 regions were detected between the two viruses [77,78]. Sequence analysis has shown that HIV and SARS-CoV proteins share common motifs that shape a degree of homology [79]. In addition, studying similar antigenic features could help in engineering super antibodies that neutralise different pathogens [80].

Evidence in the literature regarding the connection between SARS-CoV-2 and cancer at various levels remains extremely limited. Most related studies examine the immune response to SARS-CoV-2 in patients with cancer [81]. However, the cross-reactivity phenomenon between SARS-CoV-2 and malignancy-related antigens has not been previously reported. Conversely, since the COVID-19 pandemic was declared in 2020, numerous reports in the literature have linked the immune response against SARS-CoV-2 proteins with self-antigens, thus unveiling putative COVID-19 implications for autoimmune disorders including immune thrombocytopenic purpura, Guillian–Barrė syndrome and subvariants, antiphospholipid antibodies and lupus anticoagulant, Kawasaki and multisystem inflammatory syndrome in children [82,83,84,85]. In general, the emergence of autoimmunity after viral infections involving EBV, CMV, HTLV-1, herpes and hepatitis virus among others has been thoroughly described in the literature [86].

Interestingly, in the MIRA dataset there are some CD4+ and CD8+ T cell TCRs that have common CDR3 sequences. The same observation was made when comparing CDR3 sequences between MIRA and other TCR repositories. There are numerous TCRs that share the same CDR3 sequence, which originate from CD8+ T cells in MIRA and CD4+ T cells in other databases. However, the functional relevance of CD4+ and CD8+ T cells with common CDR3 sequences remains obscure, since to the best of our knowledge this phenomenon has not been debated in the literature.

The observations in this study are of multilevel significance, ranging from putative indirect protection from severe COVID-19 infection based on vaccination against and/or previous exposure to Influenza viruses, *M. tuberculosis* and other pathogens, to the association of SARS-CoV-2 related TCRs with malignancies and autoimmune disorders. However, there are several limitations related to this study. The MIRA dataset includes a very limited number of samples associated with COVID-19-acute, COVID-19-non-acute and COVID-19-exposed cohorts, thus prohibiting any statistical or ML analysis that potentially connects TCR profile irregularities with disease severity. We did however manage to make statistically significant assumptions using samples derived from COVID-19-convalescent and healthy cohorts, although ideally the number of these samples should be higher. Due to the limited number of MIRA samples, the generalisation capacity of the ML models presented here would possibly be limited on other SARS-CoV-2 TCR-Seq datasets. However, these models are serving a specific purpose in this study, which is to use them for extracting meaningful biological information related to the antigens that can distinguish MIRA cohorts. Another limitation relates to the availability of HLA allele information connecting TCR clonotypes and epitopes in McPAS, TCR3d and VDJdb. The cross-reactivity analysis focused only on the CDR3 amino acid sequence comparison between the MIRA dataset and other databases, due to the limited availability of the relevant HLA information for most TCR clonotypes in the external repositories. Therefore, it should be noted that the cross-reactions described here could only take place in individuals with specific HLA alleles, enabling the presentation of relative epitopes to potential long-lived memory T cells developed during previous infection and/or vaccination. Additionally, the extent of cross-reactivity is delimited by the inherent data bias in McPAS, TCR3d and VDJdb, stemming from the scientific community’s focus on specific pathogenic cases.

## 5. Conclusions

We believe that our study provides a novel ML-based computational framework for analysing TCR-Seq datasets and systematically highlights the breadth and depth of “cross-talk” between antigens from different pathogens, a phenomenon that may also exhibit therapeutic implications, especially in the COVID-19 context. It is our view that the scientific community should accelerate the effort of generating TCR profiling data without omitting the relevant HLA information, and include assays based on as many human pathologies as possible. ML can play a pivotal role in combining such data with multiomics and epidemiological surveillance, to build intelligent infrastructures that can be an invaluable asset in fully understanding the underlying immune response complexity.

## Figures and Tables

**Figure 1 biology-11-01531-f001:**
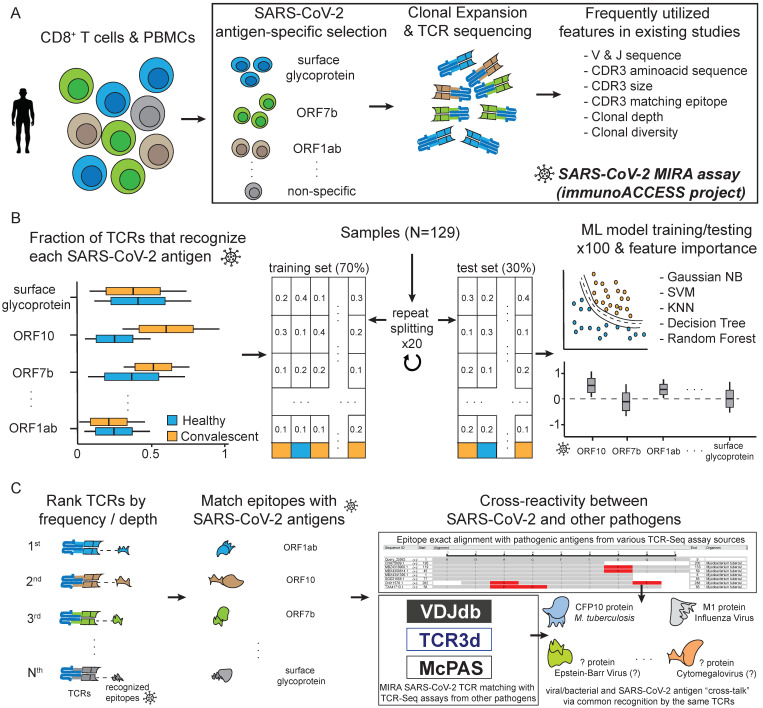
Overview of this study. (**A**) Outline of the Multiplexed Identification of T cell Receptor Antigen (MIRA) assay and the corresponding dataset available from the immunoACCESS© project web resource. (**B**) Analytic steps in this study, regarding the novel utilisation of the MIRA dataset for training Machine Learning algorithms that can highlight important SARS-CoV-2 antigens for distinguishing samples between healthy and COVID-19-convalescent cohorts. (**C**) Strategy for exploring T cell cross-reactivity between SARS-CoV-2 and other pathogens and diseases.

**Figure 2 biology-11-01531-f002:**
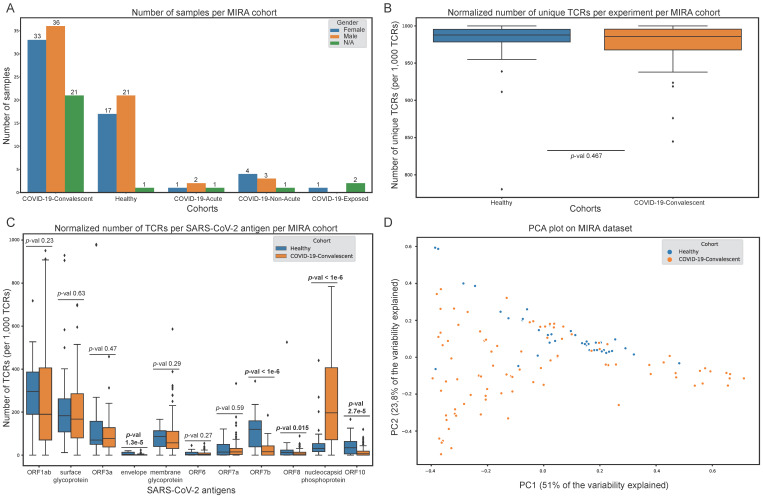
Exploratory analysis of the Multiplexed Identification of T cell Receptor Antigen (MIRA) dataset. (**A**) Number of samples in each MIRA cohort. (**B**) Per sample normalised number of unique T cell receptors (TCRs) in the healthy and convalescent cohorts. (**C**) Per sample normalised number of TCRs that recognise each SARS-CoV-2 antigen in the healthy and convalescent cohorts. (**D**) Projection of healthy and convalescent samples on the principal component analysis (PCA) space. Healthy and convalescent distributions in (**B**,**C**) were compared with the Mann-Whitney test.

**Figure 3 biology-11-01531-f003:**
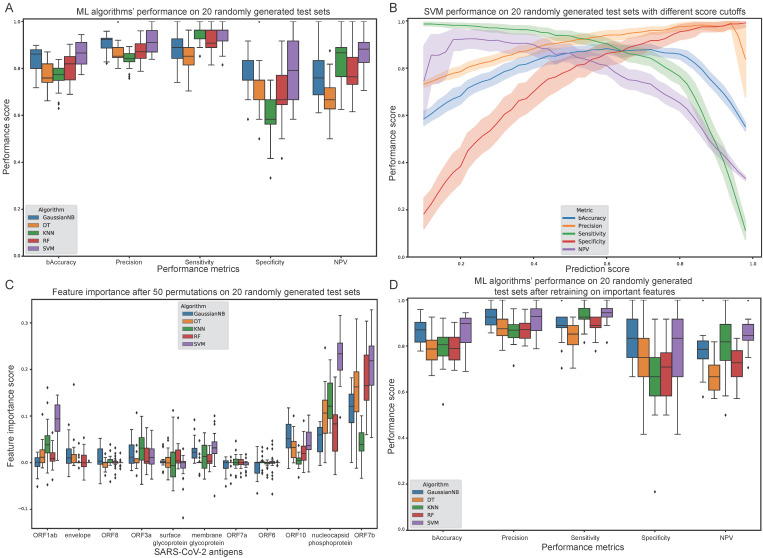
Evaluation of Machine Learning (ML) algorithms trained on the healthy and convalescent cohorts in the Multiplexed Identification of T cell Receptor Antigen (MIRA) dataset. (**A**) Balanced accuracy, precision, sensitivity, specificity and negative predictive value (NPV) of each algorithm after selecting a prediction score cut-off of 0.5. (**B**) Support Vector Machines (SVM) performance on multiple prediction score cut-offs. (**C**) Feature importance score after 50 permutations on all 20 randomly generated test sets. (**D**) ML algorithms’ performance after selecting only the important features for each algorithm and retraining.

**Figure 4 biology-11-01531-f004:**
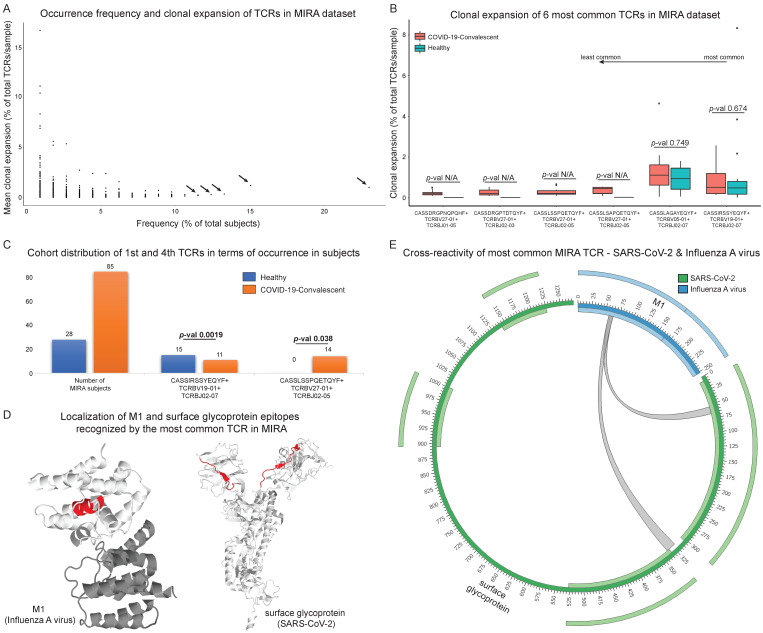
Exploration of the most common Multiplexed Identification of T cell Receptor Antigen (MIRA) T cell receptors (TCRs) in terms of clonal expansion and cross-reactivity. (**A**) Occurrence frequency and clonal expansion of all MIRA TCRs. The arrows point to the six most common TCRs (present in at least 11.5% of total number of subjects) that were further analysed in terms of clonal expansion in the two cohorts based on Mann-Whitney test (**B**). The statistical test could not be performed for some TCRs (denoted as *p*-val N/A). (**C**) Cohort distribution of the first and fourth most common MIRA TCRs that were found to be enriched in either cohort after applying Fisher’s exact test. (**D**) Secondary structure of surface glycoprotein (SARS-CoV-2) and Matrix protein 1 (M1) with cross-reactive sections, based on the most common MIRA TCR, highlighted with red colour. Locations highlighted with red colour consist of epitopes recognised by the cross-reactive TCRs and putatively reflect protein domains with similar structural or physicochemical properties. (**Ε**) Circular plot, as an alternative view of (**D**), depicting the cross-reactive property of the most common MIRA TCR that recognises epitopes from surface glycoprotein and M1. The inner and outer light-colored tracks represent the annotated domains.

**Figure 5 biology-11-01531-f005:**
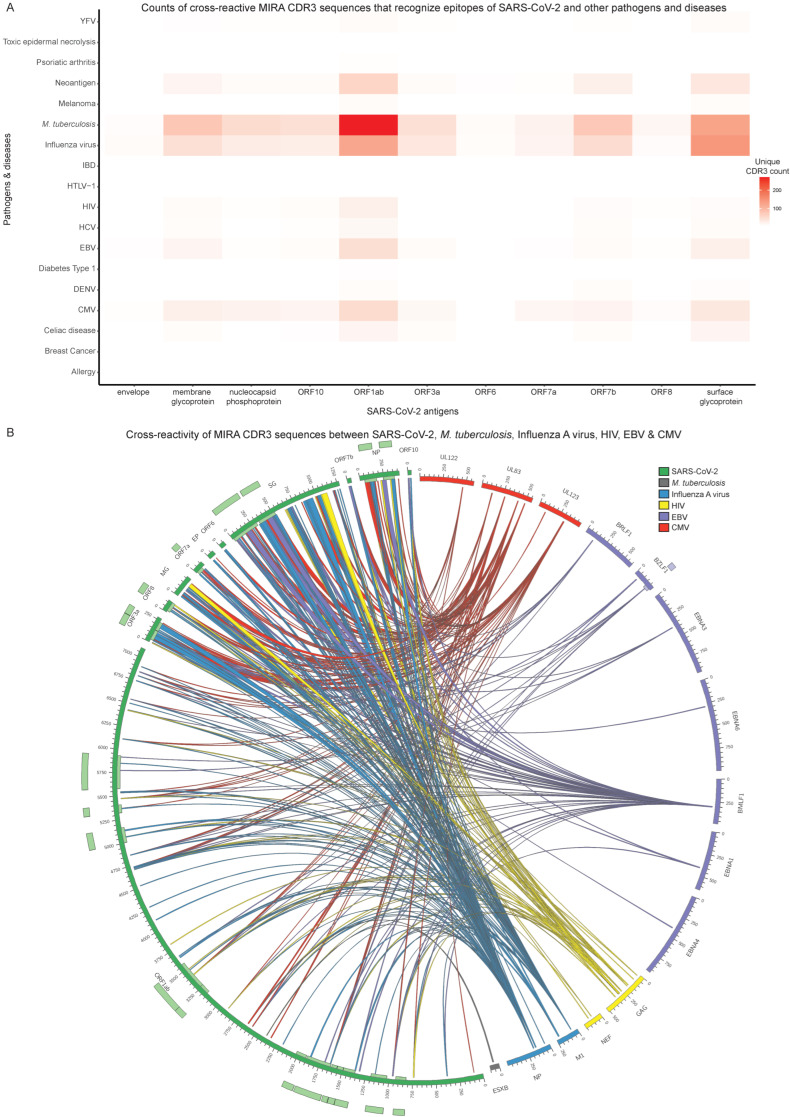
Cross-reactivity analysis of all Multiplexed Identification of T cell Receptor Antigen (MIRA) T cell receptors (TCRs). (**A**) Heatmap of unique MIRA complementarity-determining region 3 (CDR3) counts that exhibit cross-reactivity between SARS-CoV-2 (*x*-axis) and other pathogens and diseases (*y*-axis). The heatmap values correspond to the number of unique cross-reactive CDR3 sequences. (**B**) Circular plot that depicts the cross-reactivity of MIRA CDR3 regions between antigens that originate from SARS-CoV-2 and a selected subset of pathogens from (**A**). The inner and outer light-coloured tracks represent the annotated protein domains. Each connection represents the ability of a single CDR3 region to recognise a part of a SARS-CoV-2 antigen and a part of another pathogen’s protein. The connections are coloured based on their corresponding non-SARS-CoV-2 pathogens.

## Data Availability

The MIRA dataset was retrieved from the immuneACCESS© database [15]. Additional TCR CDR3 sequences from both CD4+ and CD8+ T cells were downloaded from McPAS [48] (version 4 January 2022), TCR3d [49] (version 13 January 2022) and VDJdb [50] (version 22 March 2022). All protein sequences were downloaded from UniProt [87]. The cross-reactivity exploration was achieved with custom Python scripts and Circos [88]. The three-dimensional structure of M1 and surface glycoprotein antigenic molecules was generated with Jmol, within Jalview software based on 1AA7 (A and B chain view) and 6 × 29 (A chain view) Protein Data Bank [89] entries for M1 and surface glycoprotein, respectively. The antigenic epitopes were aligned on reference sequences using the Clustal algorithm. The statistical, dimensionality reduction, ML and feature importance analyses were performed with in-house developed software based on Python’s scipy and scikit-learn as well as R’s rstudioapi, dplyr, plyr, GLDEX, TSDT, stats, stringr and ggplot2 libraries. The code and processed data used in this study can be found in gitlab.com/hyimmera/mira_analysis (accessed on 23 September 2022). Further inquiries can be directed to the corresponding authors.

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
