# Peer review of "Machine-Learning-Assisted Analysis of TCR Profiling Data Unveils Cross-Reactivity between SARS-CoV-2 and a Wide Spectrum of Pathogens and Other Diseases"

_biology, 2022, doi:10.3390/biology11101531_

Round 1
Reviewer 1 Report
This was an excellent report on very thorough research, and I approve of its publication. The literature review was thorough, the methodology was painstakingly thorough and incorporated the use of sufficient numbers of samples in exposed COVID subjects and healthy controls analysis. The codes and used input features and files have been properly named and curated avoiding any confusion. The codes were tested and found in accordance with the results mentioned in manuscript.
Author Response
We would like to thank the reviewers for their positive comments as well as the constructive criticism. In our effort to address the reviewers’ suggestions, we believe that our manuscript was improved. All changes in the manuscript were highlighted using the track changes feature of MS Word. The reviewer comments can be found below with bold, along with our point-by-point response.
Reviewer 1
English language and style
( ) Extensive editing of English language and style required
( ) Moderate English changes required
(x) English language and style are fine/minor spell check required
( ) I don't feel qualified to judge about the English language and style
Does the introduction provide sufficient background and include all relevant references?
(Yes) (Can be improved) (Must be improved) (Not applicable)
( ) (x) ( ) ( )
Are all the cited references relevant to the research?
(x) ( ) ( ) ( )
Is the research design appropriate?
(x) ( ) ( ) ( )
Are the methods adequately described?
(x) ( ) ( ) ( )
Are the results clearly presented?
(x) ( ) ( ) ( )
Are the conclusions supported by the results?
(x) ( ) ( ) ( )
Comments and Suggestions for Authors
This was an excellent report on very thorough research, and I approve of its publication. The literature review was thorough, the methodology was painstakingly thorough and incorporated the use of sufficient numbers of samples in exposed COVID subjects and healthy controls analysis. The codes and used input features and files have been properly named and curated avoiding any confusion. The codes were tested and found in accordance with the results mentioned in manuscript.
We would like to thank the reviewer for the positive comments and the thorough revision of both the manuscript and code.
Reviewer 2 Report
"Machine learning assisted analysis on TCR profiling data from COVID-19-convalescent and healthy individuals unveils cross-reactivity between SARS-CoV-2 and a wide spectrum of pathogens and other diseases" is quite suitable for the journal. But before considering for the publication following changes need to be done.
The organization of the article is well and clearly written.
The title should be shorter.
All of the shapes are unacceptably blurry. Pixels appear under the figure captions.
Figure descriptions are too long.
The programming language used in the article should be stated.
Author Response
We would like to thank the reviewers for their positive comments as well as the constructive criticism. In our effort to address the reviewers’ suggestions, we believe that our manuscript was improved. All changes in the manuscript were highlighted using the track changes feature of MS Word. The reviewer comments can be found below with bold, along with our point-by-point response.
Reviewer 2
English language and style
( ) Extensive editing of English language and style required
( ) Moderate English changes required
( ) English language and style are fine/minor spell check required
(x) I don't feel qualified to judge about the English language and style
Does the introduction provide sufficient background and include all relevant references?
(Yes) (Can be improved) (Must be improved) (Not applicable)
(x) ( ) ( ) ( )
Are all the cited references relevant to the research?
(x) ( ) ( ) ( )
Is the research design appropriate?
(x) ( ) ( ) ( )
Are the methods adequately described?
(x) ( ) ( ) ( )
Are the results clearly presented?
(x) ( ) ( ) ( )
Are the conclusions supported by the results?
(x) ( ) ( ) ( )
Comments and Suggestions for Authors
"Machine learning assisted analysis on TCR profiling data from COVID-19-convalescent and healthy individuals unveils cross-reactivity between SARS-CoV-2 and a wide spectrum of pathogens and other diseases" is quite suitable for the journal. But before considering for the publication following changes need to be done.
The organization of the article is well and clearly written.
We would like to thank the reviewer for the positive and constructive comments.
The title should be shorter.
We have changed the title to “Machine learning assisted analysis on TCR profiling data unveils cross-reactivity between SARS-CoV-2 and a wide spectrum of pathogens and other diseases”.
All of the shapes are unacceptably blurry. Pixels appear under the figure captions.
We thank the reviewer for the apt remark. The figures were created using Adobe Illustrator. However, for the purpose of the revision process we converted them to jpeg format and then inserted them to the MS Word formatted manuscript. Thus, the figures were subjected to compression twice which resulted to certain distortions. In the final submission, all figures will be uploaded in high-resolution pdf or ai format.
Figure descriptions are too long.
The description of Figures 4 and 5 were shortened without altering the conveyed message. We believe that the remaining Figures 1, 2 and 3 already have concise descriptions.
The programming language used in the article should be stated.
The programming languages and libraries as well as all external software and resources used for this study are stated in the Data Availability Statement section.